# Towards a Gaze-Driven Assistive Neck Exoskeleton via Virtual Reality Data Collection

Jordan Thompson
jordan.thompson@utah.edu
School of Computing
University of Utah
Salt Lake City, Utah, USA

Haohan Zhang
haohan.zhang@utah.edu
Dept. of Mechanical Engineering
University of Utah
Salt Lake City, Utah, USA

Daniel Brown
daniel.s.brown@utah.edu
School of Computing
University of Utah
Salt Lake City, Utah, USA

## ABSTRACT

Dropped head syndrome is an issue faced by many individuals affected by neurodegenerative diseases. This makes it impossible for these people to support their own head with their neck, causes pain and discomfort, and makes it difficult to perform everyday tasks. Our long-term goal is to use a powered neck-exoskeleton to restore natural neck motion for people with dropped head syndrome. However, determining how a user would like to move their head is challenging. We propose to leverage virtual reality as a way to collect coupled eye and head movement data from healthy individuals to train a machine learning model that can predict user-intended head movement from eye-gaze alone. We present preliminary results demonstrating the potential of our learned model. We discuss our ongoing work to compare our learned model with existing, non-learning-based methods. Finally, we discuss our future plans to incorporate human-in-the-loop feedback to enable customization of an assistive robotic neck exoskeleton for users with dropped head syndrome.

## CCS CONCEPTS

• **Human-centered computing** → **Accessibility systems and tools**; **Virtual reality**; • **Computing methodologies** → **Neural networks**; • **Computer systems organization** → **Robotic control**.

## KEYWORDS

virtual reality, neck exoskeletons, gaze detection, intent recognition

**ACM Reference Format:**
Jordan Thompson, Haohan Zhang, and Daniel Brown. . Towards a Gaze-Driven Assistive Neck Exoskeleton via Virtual Reality Data Collection. In *VAM-HRI'23: ACM International Conference on Human Robot Interaction, March 13–16, 2018, Stockholm, Sweden.* ACM, New York, NY, USA, 7 pages.

## 1 INTRODUCTION

Dropped head syndrome (DHS) [7, 16] is characterized by the inability of a person to move and raise their head. DHS results from neck muscle weakness which can arise from diverse causes, including central/peripheral neurological pathology (e.g., amyotrophic lateral sclerosis, Parkinson's disease) and autoimmune conditions (e.g., polymyositis) [3, 8, 13, 24]. People with minor or moderate DHS cannot maintain an upright head posture for extended time due to fatigue. In severe cases, the head completely drops, resulting in a "head-on-chest" posture. DHS causes pain, spinal deformity, as well as difficulty with respiratory functions, ambulation, and social interactions, which severely impacts patients' physical and emotional well-being and overall quality of life.

Static bracing of a patient's neck is a common treatment for DHS; however, these braces are often uncomfortable and ineffective [18, 20]—static braces support the head at the chin, making it difficult to speak, swallow, and breathe while wearing it. These braces also do not allow motion for daily tasks (e.g., feeding, horizontal vision, etc.). Based on our interviews with people with DHS, few patients use their prescribed neck braces at home, leaving their conditions untreated and worsening their quality of life. Other treatments for DHS include reclining wheelchairs and adding straps to a chair. These solutions, however, are not portable and also do not restore head-neck motions.

In our prior work [28–30], we invented the world's first powered neck exoskeleton, which can provide the much needed assistive neck motion for people with DHS. However, despite having the physical hardware capable of moving a patient's neck, there remains the question of how the patient would like their head to be moved and oriented. Using a hand-held device (e.g., joystick, keyboard) to control the neck exoskeleton is unintuitive and even infeasible for many patients, especially those whose DHS is a result of widespread neural degeneration (e.g., amyotrophic lateral sclerosis).

Recently, we have demonstrated the feasibility of using directional eye gazes to control the neck exoskeleton to perform a tracking task in healthy subjects [5]. This control strategy capitalizes on the fact that eye muscles are generally not affected by neural degeneration which presents an inclusive solution for people with DHS caused by neurodegenerative diseases. However, this control strategy still lacks the necessary intuitiveness because (1) the scheme is primitive and unnatural, requiring the user to use their eyes as a joystick to move their head in one of the four cardinal directions, and (2) the control parameters are fixed, precluding adaptation to an individual's behaviors and preferences. Additionally, this control strategy has only been evaluated in a user study with healthy individuals—the efficacy of using eye gaze to control the neck exoskeleton is unknown for patients with DHS. Therefore, there is a critical need to identify better models for predicting head-neck movement conditioned on natural eye gaze patterns and to develop personalization strategies for head-neck movement assistance for people with DHS.

Our long-term goal is to restore head-neck motions for people with DHS through an at-home, personalized, and easy-to-use wearable robotic solution. In this paper we describe our proposed approach to leverage virtual reality (VR) to train a machine learning

*VAM-HRI'23, March 13-16, 2023, Stockholm, Sweden*

model to predict head-neck movements given access only to a patient's gaze. We believe VR is an ideal way to collect paired head and eye movement data which will enable the development of machine learning models that learn to predict intended head movement from gaze history. Rather than requiring complex physical environments for collecting paired head and eye movement data, we argue that VR provides an inexpensive, flexible, scalable, and highly accurate method for collecting paired eye and head movement data to train our predictive models.

We hypothesize that head-neck movements can be predicted by eye movements in multiple gaze conditions (e.g., smooth pursuit, saccade) using a task-agnostic machine learning model. We test this hypothesis through validations of predictive models against ground truth head movement from a dataset collected during a pilot study with healthy adults interacting with multiple simulated virtual environments using a virtual reality headset.

## 2 RELATED WORK

Early neurophysiology studies [2, 9, 11] suggested that the movements of the head and eyes are tightly coupled during visual tasks through neural pathways like vestibulo-ocular reflex. However, the design of these studies were limited to controlled environments and tasks, as well as laboratory-bound equipment. These make it difficult to translate these early results directly to predict head-neck motions using eye movements in real-world settings. With modern wearable sensors, it becomes feasible to study gaze behaviors and head-eye coordination in real-world settings. For example, recent work collects head-free gaze data during four physical daily tasks using a wearable system [15] . However, the tasks required significant physical setup and modification costs and data collection was limited to participants who had access to the physical infrastructure. By contrast, we propose to leverage virtual reality (VR) to enable rapid environment customization and data collection without requiring large physical spaces or expensive physical setups. Furthermore, once a VR environment is built, the software can easily be shared, enabling highly scalable data collection from anyone with access to the same VR system. Our work is motivated by prior works showing that a user's attention can be detected through gaze and head movement in virtual and augmented reality settings to enable gaze-based selection and pointing [1, 22, 23, 26]; however, unlike our approach, these prior works do not predict head movement from gaze, which is necessary to enable head-neck motion directly from a user's gaze patterns.

Our paper builds upon our prior efforts to build a powered neck exoskeleton [28–30]. While the physical hardware has been completed, there remains the question of a natural interface for inferring a user's intended neck motion our prior work focuses on allowing a user to move their head (north, south, east, or west) by moving their eyes in the desired cardinal direction [5]. We seek to improve upon this work by using a recurrent neural network architecture to predict when and how a user would like their head moved based on their eye gaze history.

## 3 PROBLEM STATEMENT

Dropped head syndrome (DHS) is a common problem among individuals who experience a neurodegenerative disease. This occurs

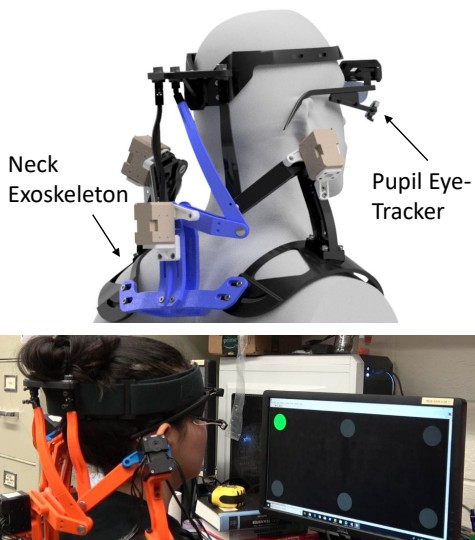

**Figure 1: (Top) A computer design model of the integration of the neck exoskeleton and a head-mounted eye-tracker. (Bottom) A healthy participant using the previous quadrant approach to control the neck exoskeleton with their eye movement [5].**

when the affected person's neck no longer has the strength to support their head. We seek to restore natural head-neck motion via a powered neck exoskeleton and eye tracking software. Given the patient's current head angle, $h_t$, at time $t$ and gaze vector, $g_t$, obtained from an eyetracker, we want to predict the user's desired head position for the next time step, $\hat{h}_{t+1}$.

## 4 METHODOLOGY

We propose a solution to this prediction problem that uses coupled head and eye movement data collected using a virtual reality (VR) headset to train a Long-term Short-term memory (LSTM) machine learning model that learns the relationship between the past history of a person's eye movement and their intended head-neck movement. Our key insight is that we can use a VR headset to easily and accurately track the head and eye movements of healthy individuals as they interact with a variety of virtual environments. This enables a highly scalable and flexible platform for collecting large amounts of coupled head and eye movement data. Using this data, we propose to train a model to predict future intended head movement given past eye movement data. Once trained, we plan to deploy this model on the actual robotic neck exoskeleton to restore natural head movement to users with DHS by tracking their eye gaze.

### 4.1 Powered Neck Exoskeleton

Static neck braces are often used to treat DHS. However, these braces are often uncomfortable and ineffective and can lead to pain and breathing problems. This is because the braces tend to apply pressure underneath the chin which forces the mouth to be

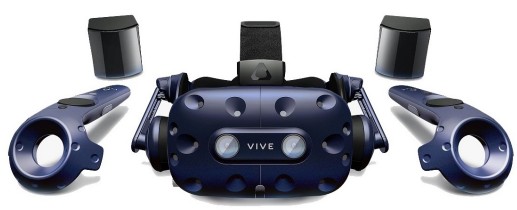

**Figure 2: An image of the Vive Pro Eye used for data collection and experimentation.**

closed. The mechanical neck brace that we propose would solve these issues by allowing for movement of the head and by applying pressure at the forehead instead of the chin (Figure 1).

## 4.2 Data Collection

Data for this project was collected in a VR environment developed in Unity using the Vive Pro Eye as shown in Figure 2. This device was used to track how the individuals moved their eyes and head together throughout a series of testing environments. These tests include rapid movement, linear smooth pursuit, and curved smooth pursuit. The data collected includes the following vectors: left eye gaze direction, right eye gaze direction, and the forward vector from device orientation representing head angle.

*4.2.1 Rapid Movement.* The Rapid Movement test is designed to monitor how an individual glances quickly in different directions around an environment. Within this test, three cubes are randomly instantiated within a $60°$ head rotation of the user. The user is instructed to look for the cubes and fixate their gaze on them for a set amount of time before the cube is re-instantiated at a new location.

*4.2.2 Linear Smooth Pursuit.* Linear Smooth Pursuit is a test of how an individual tracks an object moving in a straight line in their field of view. The user is instructed to track the object as it moves around within the environment. In this environment, a singular cube is instantiated in front of the user and moves in a straight line to a new randomly assigned point with a fixed speed.

*4.2.3 Curved Smooth Pursuit.* Curved Smooth Pursuit is very similar to Linear Smooth Pursuit with the change that now the cube moves in a randomized arc pattern. The cube then moves with a fixed angular speed along the arc. The user is instructed to track the object as it moves through the environment.

## 4.3 Simulated Gaze Training

Once the data has been collected, we train an LSTM network to predict the next head angle given the current gaze and head orientation.

$$\hat{h}_{t+1} = f_\theta(g_t, h_t) \tag{1}$$

where $g_t$ and $h_t$ are the gaze vector and the head angle respectively at time-step $t$, and $f_\theta$ is the learned model. The LSTM is trained

to minimize the mean squared error between the predicted and ground-truth head angle:

$$\mathcal{L}(\theta) = \frac{1}{T} \sum_t \|h_t - \hat{h}_t\|^2. \tag{2}$$

Figure 3 illustrates the results from when this model is trained purely on the ground truth data that was collected. What these results show is that training on this data does not lead to a model that works in practice due to a very quickly growing compounding error. The green line in figure 3 illustrates what occurs when the model is no longer given access to the ground truth head angle and is instead required to use its own output as an input (as would occur in an application of this model). We hypothesize that this is due to how a person's eye motion is dependent on their head motion. In other words, if a person's head were to be rotated too far or too little, then the resulting eye motion would account for this prediction error.

To train a more robust model, our insight is to take into account the vestibulo-ocular reflex of the eye [21]. This reflex stabilizes the eyes relative to a person's environment and compensates for head movements—this reflex is why you can move your head but keep your eyes fixed on a visual target without your gaze slipping off the target. When the head is moved, the eyes move the same distance but in the opposite direction.

To incorporate this insight into our model training, the LSTM training data is augmented with the following assumption (visualized in Figure 4): at any given time-step, an individual's gaze is fixed on a point in space regardless of how their head is oriented. In terms of vectors, this can be seen as

$$\hat{g}_t + \hat{h}_t = h_t + g_t \tag{3}$$

where $h$ and $g$ are the ground truth head and gaze vectors for time-step $t$ respectively, $\hat{h}$ is the predicted head vector for time-step $t$, and $\hat{g}$ is the simulated gaze value using the assumption that the person is looking at the same point in space at time $t$, even if we move the head using the neck exoskeleton.

Rearranging terms we can solve for the assumed human gaze given a different head position $\hat{h}_t$:

$$\hat{g}_t = h_t + g_t - \hat{h}_t. \tag{4}$$

We use this augmented gaze vector for training a robust head angle predictor. At the beginning of training, we initialize $\hat{h}_1 = f_\theta(g_0, h_0)$, but for subsequent time-step, we output predictions as

$$\hat{h}_{t+1} = f_\theta(\hat{g}_t, \hat{h}_t) \tag{5}$$

where the inputs to the model are now predicted values as opposed to ground truth data. We still minimize the mean-squared error (Equation 2). Figure 5 shows the naive and augmented training pipeline for the LSTM. Figure 6 shows the results when training under this new assumption. We can see that the model's predictions are very similar to those of the ground truth. Importantly, the outputs now account for the error of previous time-steps.

## 5 ONGOING AND FUTURE EXPERIMENTS

We are currently working on evaluating our learned model both quantitatively and qualitatively. For initial testing, we plan to take advantage of another benefit of VR, the ability to simulate and

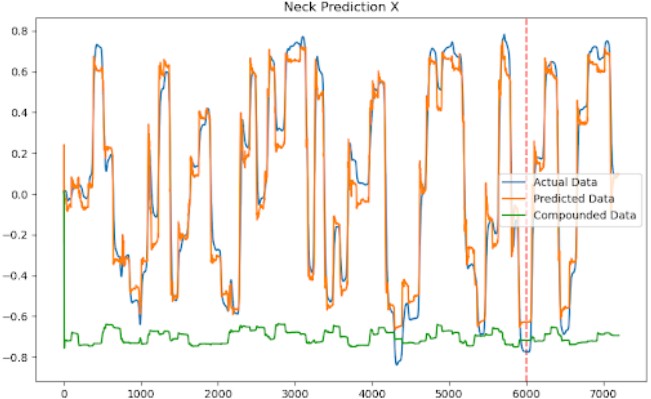

**(a) Model's predicted x component of the head angle vector when trained on ground truth data.**

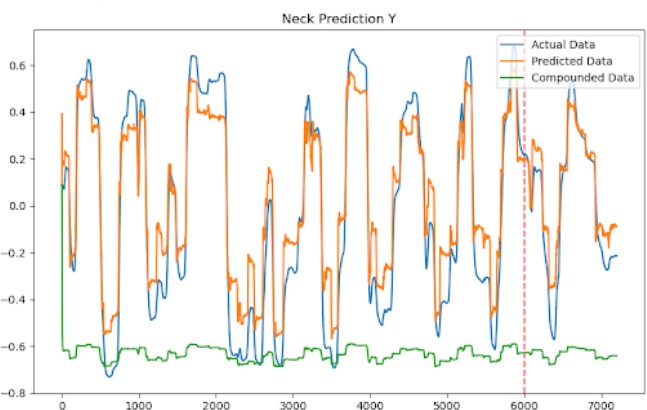

**(b) Model's predicted y component of the head angle vector when trained on ground truth data.**

**Figure 3: Results from model trained on the ground truth data collected. The blue line represents the ground truth head angle. The orange line shows the model's output when always using the ground truth inputs. The green line demonstrates what occurs when the model uses its own output for the next time-step's input. The red dashed line shows when training data ends and validation data begins.**

pilot assistive neck movements without requiring users to wear a physical neck exoskeleton. We plan to use the eye tracking in the VR headset, but ignore the user's actual head movements—we will ask the user to not move their head, but even if they do, the VR field of view will not change. Then by tracking their gaze we can predict their intended head movement using our model and use the output of our model to change the VR field of view. In this way we can safely and easily test our assitive neck movements first in virtual environments, even with healthy individuals. Then once our model works well in simulation, we can port it over to the physical robot.

In the near future, we plan to compare our gaze-conditioned, head movement prediction model with two baseline, non-learning models. The first baseline is a direct implementation of the quadrant based approach from prior work [5]. The second baseline model

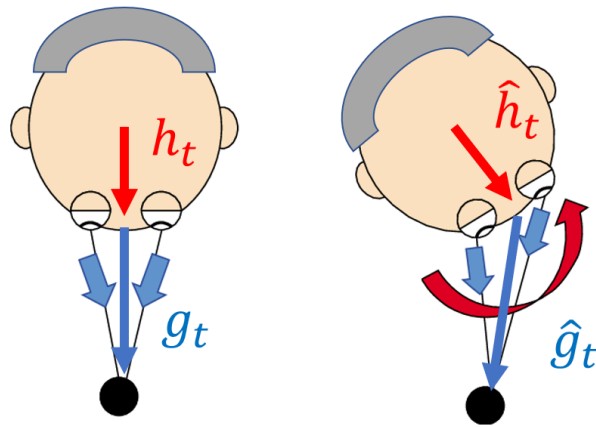

**Figure 4: We perform data augmentation on the gaze vector to enable more robust LSTM training. The augmented gaze vector $\hat{g}_t$ is based on the vestibulo-ocular reflex [21]: when looking at an object, the eyes rotate counter to head rotations to keep the object in the field of view.**

is an extension of the quadrant method that computes a vector between the current head angle vector and the gaze vector. The head is then rotated in the direction of this vector. This comparison will provide an understanding of the benefits and drawbacks of a machine learning approach versus more predefined approaches to this issue.

## 5.1 Quantitative Analysis

We propose two methods for quantitatively measuring the performance of the learned model versus the baseline methods. The first approach involves providing users with a clicker that they will use to indicate moments in which the model did not meet their preferences. The second method will be to implement a scoring metric into the simulated virtual environments to measure how well the individuals performed with each model.

*5.1.1 Preference Analysis.* Providing users with a clicker to provide feedback over each model will allow us to gain an understanding of the preferences between models for each individual. While undergoing each of the simulated virtual environments, users will be instructed to press the clicker anytime that the model does not rotate their head in a way that they found natural and comfortable. This method of analysis will provide a quantitative metric of preference over each model.

*5.1.2 Performance Analysis.* The next approach for quantitative analysis will be focused on analyzing the performance of each user within the virtual environments. This will take the form of adding a scoring metric to each environment (e.g. amount of time spent focused on a cube). This will provide an understanding of how well each model performed with respect to keeping the desired object within a reasonable field of view.

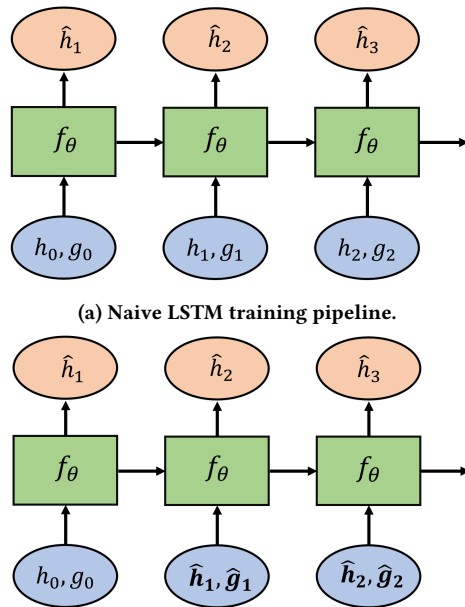

(a) Naive LSTM training pipeline.

(b) Augmented LSTM training pipeline based on the vestibulo-occular reflex.

Figure 5: A comparison of two different methods for training the LSTM to predict future head angles. In (a) the model always recieves the ground-truth head position as input. This leads to low loss, but when executed at test time (where predicted head positions become the inputs at the next timestep) leads to poor performance due to compounding errors. In (b) we address the problem of compounding errors by training the model using both the predicted head angle as input as well as using an augmented gaze vector based on the assumption that a person's gaze will stay fixed on the same point in space, even if their head had been at a different angle (Figure 4 and Equation 4).

## 5.2 Qualitative Analysis

The qualitative differences between the learned model and the baseline methods will be measured using a Likert scale survey where individuals will be asked to evaluate the comfort, speed, responsiveness, etc. of each approach. This survey will provide feedback on how each individual felt about the described methods outside of just incorrect movements measured by the quantitative analysis.

## 5.3 Testing on Physical Hardware

For this project, all testing and data collection was done in virtual reality. This is because only healthy individuals were used as participants, and healthy individuals would have a natural tendency to resist the forces being applied by the neck brace. This could lead to potentially skewed or flawed data as a result. In virtual reality, however, we can simply disable head rotation tracking and force the camera to rotate using our model regardless of how participants are rotating their heads.

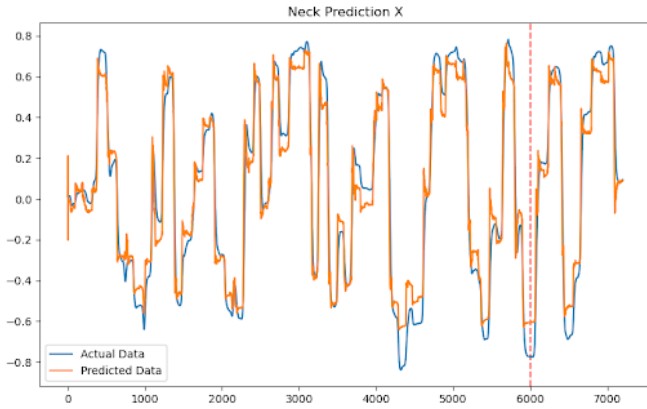

(a) Model's predicted x component of the head angle vector when trained on simulated gaze data.

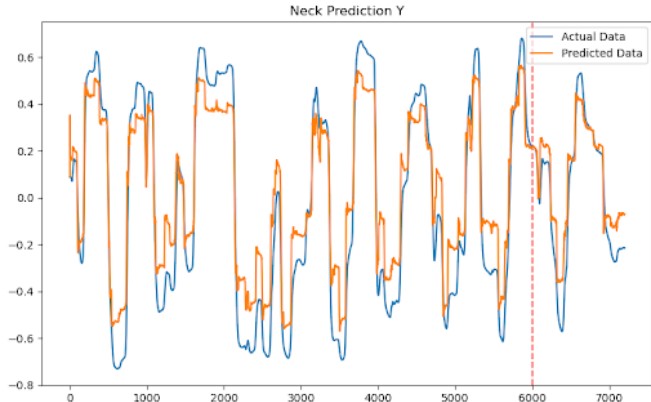

(b) Model's predicted y component of the head angle vector when trained on simulated gaze data.

Figure 6: Results from model trained using the simulated gaze assumption. The blue line represents the ground truth head angle. The orange line shows the model's output when using its prior output as its new input. The red dashed line shows when training data ends and validation data begins.

## 5.4 Expanding Virtual Reality Environments

Currently, the data for this project is being collected in trivial environments designed to measure one specific aspect of the relationship between gaze and head movements. For future development, we are expanding these environments to more realistic settings such as a driving simulator or a job simulator. This will allow us to capture the true natural movements of an individual in a dynamic environment.

## 5.5 Human-in-the-Loop Feedback and Adaptation

The previously demonstrated LSTM model may be potentially good enough to give some sense of mobility back to individuals affected by dropped head syndrome; however, Figure 6 still shows a decent

amount of error particularly around the positive and negative extremes. We also hypothesize that different individuals may have varying levels of head-eye coordination. This necessitates that the model be continually trained while in use to adapt to the user.

In the future, we plan to develop an approach that can be used for customizing continuous, gaze-controlled action (e.g., head-neck motion). Prior work has shown that human-in-the-loop feedback can enable and customize text-to-speech typing interfaces using gaze as the input [10, 17]; however, these prior works focus on tasks where finite actions (e.g., keys or words to select) are necessary. Other prior works have aimed to teach autonomous agents via sparse human feedback through user clicks [14, 27] or pairwise preference labels over a robot behavior [4, 6]. These prior works assume the human is a bystander who passively observes a robot's behavior and offers feedback.

We postulate that similar strategies can be developed to allow patients to customize the control of a neck exoskeleton while wearing the device. Prior works on adapting exoskeleton controllers have focused on either a small set of carefully selected control parameters (e.g., peak torque and its timing [12]) or objectives based on well-defined repetitive tasks (e.g., gait) using physiological signals (e.g., electromyography [19] and oxygen intake [31]). Other prior work customizes exoskeleton control by asking the user to compare two different controllers to learn their preference (e.g., during walking [25]); however, these comparisons are often difficult to make—it is often the case that two possible controllers are good and bad in different ways. To address the above technological barriers for the proposed application, we will incorporate inclusive online human feedback (through a clicker) to adapt the gaze control of the neck exoskeleton for diverse head-eye coordination behaviors (e.g., saccade, smooth pursuit, etc.).

## 5.6 Testing on Users with Dropped Head Syndrome

Individuals experiencing dropped head syndrome would be a better demographic to test on as they would be the ones to actually use an implemented version of this system. It is also probable that these individuals may have different tendencies or preferences than the healthy individuals used in this study.

## 6 CONCLUSION

This work proposes our preliminary steps towards novel method of assisting head-neck motion leveraging machine learning and virtual reality. We use virtual reality to collect paired head and eye movement data from users interaction with virutal worlds. We then train a general LSTM network to predict a user's head-neck motion given current head and eye data. Our results show that a naive implementation results in compounding errors and poor performance. To remedy this, we propose and evaluate an augmented training scheme inspired by the vestibulo-ocular reflex. Training an LSTM on this augmented data results in a dramatic increase in head movement prediction accuracy.

Given these promising preliminary results, we are excited about the many different avenues for future work and improvement. In the future, we plan to use human-in-the-loop feedback to enable our LSTM prediction model to be continually trained to conform

to user preferences with the help of a discriminator network. We hypothesize that the discriminator network could be trained on clicker feedback from the user which is necessary as users suffering from a neurodegenerative disease may not be able to provide any more feedback than a clicker. We propose a variety of experiments to both quantitatively and qualitatively measure the benefits and shortcomings of our proposed method compared to two baseline methods. This involves measuring both user preferences as well as model performance. We then describe the long-term directions of this work which includes testing on individuals experiencing dropped head syndrome with the physical neck exoskeleton in Figure 1, exploring different VR environments for data collection and model evaluation, and implementing a variety of potential human-in-the-loop feedback methods for lifelong learning.

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
