# OpenReview forum: "Towards a Gaze-Driven Assistive Neck Exoskeleton via Virtual Reality Data Collection"
_humanrobotinteraction.org/HRI/2023/Workshop/VAM-HRI — VAM-HRI 2023 Oral_

### Official Review · Program_Chairs · 2023-02-25
**Accept**

**Rating:** 9
**Confidence:** 5

**Review:**

Reviewer 1:

This paper describes a learning-based model for controlling an assistive robotic neck exoskeleton ultimately designed for people with dropped head syndrome (DHS). The model is an LSTM that predicts, given the history of user eye and head movement, the next intended head movement of the user. The key insight of their model is to incorporate the vestibulo-ocular reflex (ability to keep a steady gaze while the head moves) into the model itself. Initial data show the model to be promising but not within a goal range of acceptable error. This work is ongoing with future/current work described.


I recommend accepting this paper. It has clear implications in VAM-HRI, written well/clearly, and would make for great discussion.


Comments:
* Very well written/laid out paper and problem statement
* Although somewhat now accepted as singular, the word “data” should* be treated as plural within research as it is the plural of datum (or at least you will get the least amount of backlash from treating it as plural) https://www.britannica.com/dictionary/eb/qa/Is-Data-Singular-or-Plural-#:~:text=Technically%2C%20%22data%22%20is%20a,in%20visitors%20to%20state%20parks. , https://blog.apastyle.org/apastyle/2012/07/data-is-or-data-are.html , Example in Section 4 paragraph 1, “Using this data, we propose to train…” -> Using these data, we propose
* Could you expand on who was part of the data collection for 4.2?
* Please label your graph axes (e.g., Fig. 3 and 4)
* I personally found it a bit difficult to understand the data collection. I believe the data are being collected and this is closer to a progress update mid data collection. Possibly a diagram of where this is at within the process would be helpful.
* Can you quantify what would be an “acceptable” amount of error? In 5.5 you say there is a “decent” amount of error but that is difficult to understand/not entirely objective (or at least obtaining an estimate/aiming for some sort of goal level of error in future work would be a great addition)

Reviewer 2:
This paper uses VR to collect data for an assistive neck exoskeleton setup by training a machine learning to predict the user’s intended head pose as a function of their eye-gaze and head pose history. Overall this is an interesting and relevant problem to address, and I recommend this paper be accepted.

Feedback:
- In the problem statement, it says only the current head position is used for prediction, but it seems like the history of head poses is fed into the LSTM?
- There are no axes in Figure 3, they should be included.

---

### Decision · Program_Chairs · 2023-03-02

Accept (Oral)